# Effects of Aerobic Treadmill Training on Oxidative Stress Parameters, Metabolic Enzymes, and Histomorphometric Changes in Colon of Rats with Experimentally Induced Hyperhomocysteinemia

**DOI:** 10.3390/ijms25041946

**Published:** 2024-02-06

**Authors:** Marija Stojanović, Dušan Todorović, Kristina Gopčević, Ana Medić, Milica Labudović Borović, Sanja Despotović, Dragan Djuric

**Affiliations:** 1Institute of Medical Physiology “Richard Burian”, Faculty of Medicine, University of Belgrade, 11000 Belgrade, Serbia; mrj.stojanovic@gmail.com (M.S.); t.dusan@hotmail.com (D.T.); 2Institute of Chemistry in Medicine “Petar Matavulj”, Faculty of Medicine, University of Belgrade, 11000 Belgrade, Serbia; kristinagopcevic@yahoo.com (K.G.); medicana89@gmail.com (A.M.); 3Institute of Histology and Embryology “Aleksandar Ð. Kostić”, Faculty of Medicine, University of Belgrade, 11000 Belgrade, Serbia; sborovic2001@yahoo.com (M.L.B.); sanjadesp@gmail.com (S.D.)

**Keywords:** colon, hyperhomocysteinemia, lactate dehydrogenase, malate dehydrogenase, oxidative stress, rat, treadmill training

## Abstract

The aim of this study was to investigate the effects of aerobic treadmill training regimen of four weeks duration on oxidative stress parameters, metabolic enzymes, and histomorphometric changes in the colon of hyperhomocysteinemic rats. Male Wistar albino rats were divided into four groups (*n* = 10, per group): C, 0.9% NaCl 0.2 mL/day subcutaneous injection (s.c.) 2x/day; H, homocysteine 0.45 µmol/g b.w./day s.c. 2x/day; CPA, saline (0.9% NaCl 0.2 mL/day s.c. 2x/day) and an aerobic treadmill training program; and HPA, homocysteine (0.45 µmol/g b.w./day s.c. 2x/day) and an aerobic treadmill training program. The HPA group had an increased level of malondialdehyde (5.568 ± 0.872 μmol/mg protein, *p* = 0.0128 vs. CPA (3.080 ± 0.887 μmol/mg protein)), catalase activity (3.195 ± 0.533 U/mg protein, *p* < 0.0001 vs. C (1.467 ± 0.501 U/mg protein), *p* = 0.0012 vs. H (1.955 ± 0.293 U/mg protein), and *p* = 0.0003 vs. CPA (1.789 ± 0.256 U/mg protein)), and total superoxide dismutase activity (9.857 ± 1.566 U/mg protein, *p* < 0.0001 vs. C (6.738 ± 0.339 U/mg protein), *p* < 0.0001 vs. H (6.015 ± 0.424 U/mg protein), and *p* < 0.0001 vs. CPA (5.172 ± 0.284 U/mg protein)) were detected in the rat colon. In the HPA group, higher activities of lactate dehydrogenase (2.675 ± 1.364 mU/mg protein) were detected in comparison to the CPA group (1.198 ± 0.217 mU/mg protein, *p* = 0.0234) and higher activities of malate dehydrogenase (9.962 (5.752–10.220) mU/mg protein) were detected in comparison to the CPA group (4.727 (4.562–5.299) mU/mg protein, *p* = 0.0385). Subchronic treadmill training in the rats with hyperhomocysteinemia triggers the colon tissue antioxidant response (by increasing the activities of superoxide dismutase and catalase) and elicits an increase in metabolic enzyme activities (lactate dehydrogenase and malate dehydrogenase). This study offers a comprehensive assessment of the effects of aerobic exercise on colonic tissues in a rat model of hyperhomocysteinemia, evaluating a range of biological indicators including antioxidant enzyme activity, metabolic enzyme activity, and morphometric parameters, which suggested that exercise may confer protective effects at both the physiological and morphological levels.

## 1. Introduction

Homocysteine (Hcy) is a sulfur-containing intermediate product of methionine, which is metabolized in the body by remethylation and trans-sulfuration [1]. Serum Hcy levels are largely regulated by gene polymorphisms of key enzymes involved in its metabolic pathway. Also, the nutritional intake of vitamins involved in Hcy metabolism as important cofactors, such as vitamin B6, vitamin B12, and folic acid, plays a significant role in the regulation of Hcy levels [2]. In fact, low dietary folate intake is considered one of the most common causes of hyperhomocysteinemia (HHcy)—elevated levels of Hcy in the blood [3]. Hyperhomocysteinemia is defined as an Hcy concentration in the plasma higher than 15 μmol/L [3]. It has been shown that elevated Hcy levels are associated with an increased risk of cardiovascular, neurodegenerative, or urinary system disorders [4,5,6]. Furthermore, Hcy is recognized as a risk marker for cardiovascular disease development [7].

On the other hand, it has been shown that elevated Hcy levels are associated with gastrointestinal system disorders such as inflammatory bowel disease (IBD), colorectal cancer (CRC), and motility impairments [8,9,10]. In recent times, particularly great attention has been paid to Hcy as an important factor in the pathophysiology of inflammatory bowel disease, a chronic inflammatory condition that includes two entities: Crohn’s disease and ulcerative colitis. In research conducted on IBD patients, elevated Hcy levels were shown both in the blood and in the colon wall [11]. The potential cellular mechanisms through which Hcy in elevated concentrations causes tissue damage in IBD include oxidative stress, proinflammatory actions, and endoplasmic reticulum stress [12,13,14]. It has been shown that HHcy leads to oxidative stress, both by increasing the generation of free radicals with its pro-oxidant properties, and by inhibiting the antioxidant protection system. The proposed mechanisms include the autoxidation of Hcy, formation of Hcy thiolactone as a toxic intermediate of Hcy, and inhibition of glutathione peroxidase activity [15]. Moreover, recent studies reported the connection between plasma Hcy levels and the risk of developing thromboembolic complications in IBD patients [16]. This is particularly important since hypercoagulable conditions in patients with IBD represent a significant cause of morbidity and mortality. Additionally, HHcy has been also recognized as a contributing factor that promotes the progression of IBD into colorectal cancer [9].

Regardless of the numerous factors that can contribute to colorectal cancer development, it has been shown that HHcy may be one of the initiation factors, independent of the presence of IBD. This is mostly explained by defects in the functioning of key enzymes of Hcy metabolism, which use folic acid as their cofactor [17]. Oxidative stress and physical inactivity are among the other etiopathogenetic factors for the development of CRC. Numerous epidemiological studies have mainly talked about the preventive effects of physical activity on the development of colorectal cancer, although there is a lack of data on the exact molecular events at the cellular level that are responsible for reducing the risk of CRC. Physical activity is considered to have a beneficial effect on the antioxidant protection system in numerous tissues. Furthermore, experimental studies have demonstrated decreased levels of inflammatory markers in the colon of mice following physical activity [18]. Lifestyle, especially the level of physical activity, is recognized as a very important factor in this context, and potential strategies for the prevention of colorectal cancer could be aimed at modulating the level of physical activity.

Regular physical activity is one of the basic elements of a healthy lifestyle that reduces the risk for developing cardiovascular, endocrine, and musculoskeletal system disorders, as well as reducing the initiation of malignant transformation of cells [19,20]. The effects that physical activity will have on an organ system largely depend on the type, duration, and modality of the physical activity. From the perspective of the gastrointestinal system, physical activity achieves benefits such as a reduced incidence of colorectal cancer, diverticulitis, cholelithiasis, and constipation [21]. In addition, the consequences of intense physical activity such as nausea, vomiting, diarrhea, or intestinal bleeding should be mentioned. The putative mechanisms underlying gastrointestinal distress in response to physical activity include an increased production of free radicals such as reactive oxygen species (ROS) and reactive nitrogen species (RNS), resulting in oxidative and nitrosative stress [22].

As it is a known fact that the concentration of malondialdehyde (MDA) is a marker of lipid peroxidation and catalase (CAT) and superoxide dismutase (SOD) are antioxidative enzymes responsible for the removal of ROS such as superoxide anions and hydrogen peroxide, the aim of this study was to examine the effects of aerobic treadmill training on oxidative stress parameters, the activity of certain metabolic enzymes, and histomorphometric parameters of the colon of rats with experimentally induced HHcy.

## 2. Results

### 2.1. Oxidative Stress Parameters in Rat Colon Tissue

The concentration of MDA in the colon tissue was significantly lower in the CPA group (3.080 ± 0.887 μmol/mg protein) compared to the control (5.534 ± 0.798 μmol/mg protein, *p* = 0.0141). In the HPA group, the measured concentration of MDA was significantly higher (5.568 ± 0.872 μmol/mg protein) in comparison to the CPA group (*p* = 0.0128) (Figure 1A).

The activity of CAT in the colon tissue was significantly increased in the HPA group (3.195 ± 0.533 U/mg protein) compared to all other groups (C: 1.467 ± 0.501 U/mg protein, *p* < 0.0001; H: 1.955 ± 0.293 U/mg protein, *p* = 0.0012; and CPA: 1.789 ± 0.256 U/mg protein, *p* = 0.0003) (Figure 1B).

The total SOD activity in the colon tissue was significantly increased in the HPA group (9.857 ± 1.566 U/mg protein) in comparison to the C (6.738 ± 0.339 U/mg protein, *p* < 0.0001), H (6.015 ± 0.424 U/mg protein, *p* < 0.0001), and CPA groups (5.172 ± 0.284 U/mg protein, *p* < 0.0001). On the other hand, the total SOD activity was significantly lower in the CPA group compared to the C group (*p* = 0.0425) (Figure 1C).

Native electrophoresis of the SOD enzyme in rat colon tissues showed the expression of only one isoform (Mn-SOD) in the C, H and CPA groups, while in the HPA group, besides the Mn-SOD isoform, Cu/Zn-SOD was also detected (Figure 2).

### 2.2. Metabolic Enzyme Activities in Rat Colon Tissues

The total lactate dehydrogenase (LDH) activity was significantly increased in the HPA group (2.675 ± 1.364 mU/mg protein) compared to the CPA group (1.198 ± 0.217 mU/mg protein, *p* = 0.0234) (Figure 3A). Likewise, the total malate dehydrogenase (MDH) activity was significantly higher in the HPA group (9.962 (5.752–10.220) mU/mg protein) in comparison to the CPA group (4.727 (4.562–5.299) mU/mg protein, *p* = 0.0385) (Figure 3B).

Native electrophoresis of the LDH enzyme in rat colon tissues showed the expression of four different LDH isoforms: LDH 1, LDH 2 LDH 3, and LDH 4. (Figure 4A); the densitometric analysis showed differences in the level of relative activity of these isoforms between the different groups (Figure 4B). The relative activity of isoform LDH 1 was significantly higher in the HPA group (122.5 (59.05–199.20) px/mg protein) in comparison to the C (14.50 (10.40–18.75) px/mg protein, *p* = 0.0037) and H groups (15.20 (14.10–26.95) px/mg protein, *p* = 0.0327) (Figure 5A). The relative activity of the LDH 2 isoform was also significantly increased in the HPA group (336.20 ± 207.10 px/mg protein) compared to the C (122.60 ± 30.94 px/mg protein, *p* = 0.0327) and H groups (116.90 ± 56.81 px/mg protein, *p* = 0.0279) (Figure 5B). No significant differences in the relative activities of LDH 3 and LDH 4 isoforms were detected between groups (Figure 5C,D) but the values of these isoforms were highest in the HPA group.

Native electrophoresis of the MDH enzyme in rat colon tissues showed the expression of only one isoform, mitochondrial MDH (mMDH), in all groups (Figure 6).

### 2.3. Histomorphometric Parameters in Rat Colon Wall

In the physically active groups, microscopic sections of the colon tissue revealed a normal architecture, without indications of inflammatory bowel disease. Signs of cryptitis or crypt abscesses were not observed. Erosions, ulcerations, and tumorous changes in the epithelium were not evident (Figure 7).

The homocysteine load (H group) led to a statistically significant increase in the depth of the crypts (384.70 ± 18.50 μm) compared to the C group (330.90 ± 16.89 μm, *p* = 0.0020)—an increase of 16.3%. After four weeks of aerobic treadmill training, in the CPA group, the depth of the crypts decreased by 21.7% (301.10 ± 12.35 μm) compared to the H group (*p* < 0.0001). In the HPA group, the depth of the crypts decreased by 14.1% (330.60 ± 21.78 μm) compared to the H group (*p* = 0.0019) (Figure 8A).

Loading with homocysteine did not lead to a statistically significant increase in the thickness of the lamina muscularis mucosae in the H group (20.18 ± 2.02 μm) compared to the C group (17.80 ± 1.48 μm). In the control group treated with aerobic treadmill training (CPA group), the thickness of the lamina muscularis mucosae decreased by 8% (18.40 ± 1.36 μm) compared to the H group, but without reaching statistical significance. In the HPA group, there was also a significant decrease in the thickness of the lamina muscularis mucosae of the colon (18.36 ± 1.40 μm) compared to the H group (Figure 8B).

In the H group, the thickness of the tunica mucosae (393.20 ± 10.51 μm) increased by 15.1% compared to the C group (341.50 ± 26.29 μm, *p* = 0.0207). In the CPA group, the thickness of the tunica mucosae (328.00 ± 18.22 μm) decreased by 16.6% compared to the H group (*p* = 0.0037) (Figure 8C).

Although there are statistically significant changes in depth of the crypts and the thickness of the tunica mucosae, no statistical significance was measured in the total colon wall thickness between the examined groups (Figure 8D).

## 3. Discussion

In the present study, we investigated the influence of aerobic treadmill training in the experimentally induced hyperhomocysteinemia rats on oxidative stress parameters, metabolic enzyme activities, and histological changes in the rat colon.

Numerous experimental studies have investigated the connection between physical activity and the occurrence of oxidative stress [23,24,25]. The results of different studies cannot be generalized since they depend a lot on the type of experimental protocol. The extent to which physical activity affects oxidative status largely depends on the intensity and duration of the physical activity. Our research included a model of aerobic physical activity using a treadmill for small laboratory animals. According to our results, in the group that performed physical activity (CPA group), there was a decrease in the MDA concentration compared to the control group, indicating a reduced level of oxidative tissue damage after physical activity, independent of HHcy. In addition, we showed that physical activity in HHcy conditions led to an increase in antioxidant enzyme (SOD and CAT) activities in comparison to the H group. This could be explained in part as the antioxidant capacity of physical activity or as a physiological adaptive response to the ROS production induced by physical activity on a treadmill. Adaptation in this context refers to the stimulation of antioxidant enzyme production, resulting in their increased activity. Similar results were observed in one of our previous studies, which included the same methodological approach and training program on treadmills, where the oxidative stress parameters were determined in the heart tissue homogenates [26]. These findings showed elevated SOD and CAT activities in the active hyperhomocysteinemic group compared to the other groups [26]. Additionally, using a native electrophoresis method, in the present study, two isoforms of SOD (Mn-SOD and Cu/Zn-SOD) were detected on representative zymograms, with both isoforms expressed in the HPA group. This result is consistent with our previously published data, where Mn-SOD and Cu/Zn-SOD isoforms were also detected on representative zymograms and densitometric curves of rat cardiac tissue [26]. We obtained similar results using different tissue samples (heart and colon) regarding the oxidative status, although, according to literature data, it is not expect that different tissues will react in the same way to oxidative stress. The reaction largely depends on the type of tissue and the antioxidant capacity of that particular tissue [27]. Thus, research on models using acute aerobic training regimens has shown that the oxidative stress following physical activity is tissue specific [28]. It is important to mention that to a large extent, the duration of training affects the level of oxidative tissue damage. Therefore, in a study where a chronic swimming training protocol was applied in rats for eight weeks, after four weeks of treatment, the MDA concentration increased while after sixth weeks of treatment, a decrease in the MDA concentration was observed, supporting our previously mentioned concept of an adaptive response to oxidative stress during physical activity [29]. The data from the literature show conflicting results about effects of Hcy on MDA levels. There are results showing increased Hcy levels caused by a methionine load, without an increase in MDA levels [30]. On the other hand, Hcy was significantly higher in men than in women and in older versus younger subjects; at the same time, MDA levels were significantly greater only in the young men, showing a positive correlation with Hcy levels [30]. Superoxide anions are predominantly generated in cells through the activity of enzymes such as nicotinamide adenine dinucleotide phosphate (NADPH) oxidase, xanthine oxidase, and the mitochondrial electron transport chain [31]. Hcy has been observed to increase the expression of NADPH oxidase in mice with hyperhomocysteinemia, thereby contributing to an elevated production of superoxide anions [32]. The gene encoding NADPH oxidase is part of the NOX family of genes. The expression of NOX2, predominantly expressed in endothelial cells, has shown a positive correlation with Hcy concentration in human endothelial cells and rat cardiomyocytes [33,34].

Since our experimental protocol led to an alteration in the oxidative stress parameters, showing an increase in the levels of the lipid peroxidation marker MDA after four weeks of treadmill training in the HPA group compared to the CPA group, it was also of interest to examine whether these changes also affected the microarchitecture of the colon tissue samples. With that aim, we examined the following histomorphological parameters in the isolated colon segments: the depth of the crypts, the thickness of the lamina muscularis mucosae, the thickness of the tunica mucosa, and the total colon wall thickness. In the sedentary groups, there was a statistically significant increase in the crypt depth in the Hcy-treated group (H group) compared to the control. After four weeks of training, in the hyperhomocysteinemic active group (HPA), the depth of the crypts was significantly decreased compared to the H group. These data suggest that the morphometry changes detected could be, at least in part, explained by hyperhomocysteinemia. Hyperhomocysteinemia has been suggested to be a mediator of the inflammatory response in a rat experimental model of colitis induced by acetic acid [35]. The results from this study showed significant tissue damage and immune cell infiltration in all layers of the colon wall, indicating that HHcy is a significant toxic factor for colon tissues. Based on this, we can assume that the changes in the depth of the crypts in the sedentary group treated with Hcy occurred due to Hcy-induced inflammation, the accumulation of inflammatory cells, and the consequent the loss of the normal microarchitecture of the crypts. On the other hand, the decrease in the depth of the crypts in the hyperhomocysteinemic active group compared to the H group could be due to the potential anti-inflammatory effects of physical activity. Additional studies are certainly needed to confirm these assumptions. Furthermore, the histological analysis did not show statistically significant changes in the thickness of the lamina muscularis mucosae or the total colon wall thickness. The results of our previous histomorphometric studies under methionine loading conditions in rat colon tissues showed a similar trend, with a statistically significant increase in the depth of the crypts and mucosal length in the methionine-treated group compared to the control group [36]. In addition, it was shown that under methionine loading, there was an increase in the number of cells in the lamina propria, indicating a potential pro-inflammatory effect of methionine.

The experimental design of our study included an HHcy model with aerobic treadmill training. It is important to point out that this protocol, according to the literature data, corresponds to aerobic training, because at a treadmill speed of 20 m/min, it reaches a balance point where the rate of lactate production is equal to the speed of its removal, i.e., maximum stabilization of the lactate concentration in the blood of rats is achieved [37]. To determine whether there were any changes in the energy metabolism of the colon tissue, we examined the activity of the enzymes LDH and MDH. Both enzymes have important metabolic roles. LDH is an enzyme of anaerobic metabolism, while MDH catalyzes one of the Krebs cycle reactions and can be considered an aerobic metabolism enzyme [38,39]. The results of our study showed a statistically significant increase in the total activities of LDH and MDH in the HPA group compared to the active control group (CPA), indicating an effect of HHcy on the total activity of these enzymes. It is not known how HHcy led to a change in the activities of these enzymes in the rat colon tissue samples. So far, the effects of HHcy on the gastrointestinal system have been described in both human and experimental studies [40,41,42,43]. In a culture of lymphocytes isolated from the lamina propria of the rat colon, the effects of different concentrations of Hcy on the expression of inflammatory markers were examined. The results of this study showed increased expression of IL-17 in the HHcy groups in a dose-dependent manner, indicating the pro-inflammatory effects of HHcy [13]. It has been proven that these pro-inflammatory effects are based on a change in mitochondrial function in terms of increasing the activity of the respiratory chain [44]. It is also possible that in our study, HHcy with its pro-inflammatory properties leads to a change in the activity of mitochondria and consequently affects the activities of the metabolic enzymes LDH and MDH. Additionally, it is possible that HHcy with its pro-oxidative properties can cause changes in the functional groups of enzymes, thereby increasing their activity. In accordance with the results of the current study, our previous results showed a significant increase in the total LDH activity in the HPA group in comparison to the other groups in rat heart tissue homogenates [45]. Taking all this into consideration, our results showed an increase in LDH activity in both colon and heart tissues samples. Considering that LDH is associated with tissue damage, these changes can be explained by the disruption of the colon tissue structure during aerobic training in experimentally induced HHcy rats [46].

Furthermore, the electrophoresis analysis detected four isoforms of LDH (LDH 1, LDH 2, LDH 3, and LDH 4) of which, the relative activities of LDH 1 and LDH 2 showed a statistically significant increase in the HPA group compared to the H group. This change in the activity of certain isoforms could be explained by the effects of physical activity, but additional studies are needed to determine the exact mechanism.

The results by Buehlmeyer et al. show that a 12-week exercise period can downregulate betaine homocysteine methyltransferase 2 (BHMT2) in rat colon mucosa [47]. Elevated levels of DNA methylation have been recognized as a prevalent occurrence in the progression of cancer [48]. This phenomenon is associated with BHMT2, which plays a crucial role in the production of dimethylglycine and methionine [49]. Methionine metabolism is significantly implicated in DNA dynamics, serving as a vital methyl group donor. This is essential for both DNA synthesis and repair, as well as in the realm of epigenetic modifications. The generation of the methyl donor S-adenosyl methionine is facilitated through two primary pathways: a folate-dependent route utilizing homocysteine methyltransferase and the betaine homocysteine methyltransferase (BHMT) reaction. Research in the field of nutrition has indicated a heightened activity of the BHMT pathway in rodents compared to humans [50], suggesting a potential adaptive response to physical exercise, which could potentially mitigate DNA damage.

According to the findings by Demarzo et al., a decrease in the colonic expression of the cyclooxygenase-2 (COX-2) enzyme is present in physically active rats [51]. The process of colonic carcinogenesis is closely associated with an augmented expression of COX-2 and this upregulation results in increased synthesis of prostaglandin E2 (PGE2), which establishes a stimulatory feedback loop influencing various biological functions, such as an enhancement of cellular proliferation [52]. Increased levels of PGE2 in the colonic mucosa have been documented in rodent models of colon cancer [53], as well as in humans afflicted with colorectal polyps or cancer [54]. Furthermore, nuclear factor kappa B (NF-κB), one of the main signal transduction pathways sensitive to oxidative stress in mammalian tissues, warrants attention. Activation of the NF-κB cascade has been demonstrated to amplify the gene expression of critical enzymes, such as mitochondrial SOD. This augmentation plays a vital role in sustaining the cellular oxidant–antioxidant equilibrium, particularly during exercise [55]. Intriguingly, recent studies have indicated that regular exercise engenders adaptive responses, including the attenuation of NF-κB activation [56]. The expression of COX-2 is known to be stimulated by NF-κB, suggesting a potential mechanism through which exercise may exert a modulatory effect.

To explain the effects of exercise on oxidative stress, metabolism, and proliferation in colon tissues, future studies should aim to assess the differences in key signaling pathways, such as the PI3K/Akt/mTOR and MAPK pathways, which are known to play critical roles in colon cancer progression, and the use of advanced metabolomic techniques to analyze colon tissues from exercised and sedentary rats could unveil alterations in metabolic pathways due to exercise. A particular focus could be on glycolysis, fatty acid oxidation, and amino acid metabolism. An investigation of exercise-induced epigenetic changes in colon cells could be crucial to understanding how exercise influences epigenetic modifications (like the SAM/SAH ratio, DNA and RNA methylation, and histone modifications) in colon cells. Also, future research should focus on identifying and characterizing the key signaling molecules (like AMP-activated protein kinase and NF-κB) that mediate the effects of exercise on reducing Hcy-induced oxidative stress. Inhibitors or activators of these pathways could be used to delineate their specific roles.

## 4. Materials and Methods

### 4.1. Experimental Materials and Study Design

In this study, male Wistar albino rats were used as experimental animals, and their body mass was approximately 160 g at the starting point of the experimental protocol. The experimental animals were properly acclimatized and maintained in optimal laboratory conditions: stable temperature of 20 ± 2 °C, relative air humidity controlled at 50 ± 15%, and a consistent 12/12 h light/dark cycle, with the light phase starting at 07:30 a.m. The rats were housed in transparent plexiglass cages with the floor covered with woodchips and were fed with commercial rat pellets and had tap water available for drinking ad libitum.

The rats were acclimatized to the laboratory conditions for 3 days. The experimental protocol was conducted during 28 consecutive days. Rats were randomly divided into four groups (*n* = 10, per group): the C group was treated with saline (0.9% NaCl; 0.2 mL/day s.c. 2x/day) for 2 weeks; the H group was treated with Hcy (0.45 µmol/g b.w./day s.c. 2x/day [57]) for 2 weeks; the CPA group was treated with saline (0.9% NaCl; 0.2 mL/day s.c. 2x/day) for 2 weeks and aerobic treadmill training for 4 weeks; and the HPA group was treated with Hcy (0.45 µmol/g b.w./day s.c. 2x/day) for 2 weeks and aerobic treadmill training for 4 weeks. Saline and Hcy were administered subcutaneously beneath the back skin, with applications occurring twice daily at eight-hour intervals throughout the initial two-week period of the experimental protocol [57].

Rats within the active groups were subjected to an aerobic treadmill exercise regimen on specialized treadmill equipment (Lab Animal-Treadmill, Elunit, Belgrade, Serbia), which was specifically designed for the physical exertion of small laboratory animals. Rats from the active groups underwent a preparatory phase for the aerobic treadmill program over a span of three days. The rats were adapted for the protocol by running at a speed of 10 m/min, without an incline, for 10 min each day. After adaptation, the animals in the active groups ran on the treadmill apparatus for 30 min/day for 4 weeks. The speed of the belt was 20 m/min with no incline. The experiments conducted on rats by Contarteze et al. [58] has validated that at this speed, the concentration of lactate in the blood does not increase beyond a certain point during sustained training sessions. This observation suggests that if rats run on a treadmill at this speed, they will be in the zone of aerobic metabolism.

Two weeks post initiation of our experimental protocol, rats from the C and H groups were sacrificed, and after a full four weeks, the animals from the trained groups (CPA and HPA) were sacrificed by decapitation. Subsequent to this, segments of the rat colon were carefully surgically removed. The colon samples from 5 rats in each group were used for the biochemical analysis, while the colon samples from the other 5 rats in each group were used for the histomorphometric analysis. Colon samples for the biochemical analysis were homogenized in TRIC-HCl buffer in a tissue/buffer ratio of 1:10. This homogenization buffer was comprised of the following components: 20 mmol/L Tris-HCl, pH 7.5, 1% Triton X-100, 1 μg/mL leupeptin, 1 mmol/L protease inhibitor phenylmethylsulphonyl fluoride, and 250 mmol/L sucrose [59]. Following homogenization, the colon tissue samples were centrifuged for 10 min at a velocity of 10,000 rpm; thereafter, the protein concentration in the supernatant was quantified using the Bradford method [60]. The obtained supernatant samples were kept at −20 °C until further biochemical analyses of oxidative stress parameters and metabolic enzymes activities. The other colon segments were put in 4% neutral buffered formaldehyde and were fixed for the preparation of histological slides.

### 4.2. Chemicals

In this research, all chemicals were obtained from Sigma-Aldrich Chemie GmbH, Schnelldorf, Germany.

### 4.3. Determination of Oxidative Stress Parameters in Rat Colon Tissues

#### 4.3.1. Determination of Lipid Peroxidation

The assessment of lipid peroxidation was carried out by quantifying the MDA levels in the rat colon tissue homogenates using the thiobarbituric acid reactive substance (TBARS) assay [61]. A 10% solution of sodium dodecyl sulfate (SDS) was added to the tissue homogenates in a ratio of 1:1. Subsequently, a reagent containing thiobarbituric acid (7 mmol/L) dissolved in 20% aqueous acetic acid solution and a NaOH solution (1 mol/L) were added to the reagent mixture. The mixture was then incubated at 100 °C, and after 30 min, the reaction was stopped by cooling the samples on ice. After 10 min of incubation on ice, the samples were centrifuged at 4000 rpm for 5 min to separate the denatured proteins. The absorbance of the supernatants was measured at 532 nm. The concentration of MDA in the tissue homogenates was determined using the Beer–Lambert- law, dividing the absorbance by the molar extinction coefficient of the chromogenic product (x = 1.56 × 10^5^ mol^−1^cm^−1^), which was expressed in units of µmol/mg of protein.

#### 4.3.2. Determination of CAT Activity

The CAT activity in the homogenates of rat colon tissues was measured using spectrophotometry and the method of Beers and Sizer [62]. The change in absorbance was measured at 240 nm for one minute during the decomposition of hydrogen peroxide to water and oxygen by CAT. One unit of CAT activity (U) was expressed as the activity of the enzyme in the tissue sample that was able to breakdown 1 μmol hydrogen peroxide in one minute. The total activities of CAT were measured on a UV-2600 spectrophotometer (Shimadzu, Kyoto, Japan).

#### 4.3.3. Determination of Total SOD Activity

To determine the activity of SOD in rat colon tissue homogenates, the spectrophotometric method of Misra and Fridovich was used [63]. SOD has the ability to inhibit the autoxidation of adrenaline to adrenochrome. A unit of SOD activity (U) was considered the level of SOD activity in the tissue homogenate that was able to prevent 50% of the adrenaline autoxidation in the reagent mixture at a temperature of 26 °C and pH of 10.5. Changes in absorbances were measured at 480 nm. The total activities of SOD were measured using a UV-2600 spectrophotometer (Shimadzu, Kyoto, Japan).

#### 4.3.4. Determination of the SOD Isoforms

Isoforms of SOD were separated using native polyacrylamide gel electrophoresis (on a 7.5% gel) using a running buffer composed of 0.025 mmol/L Tris and 0.192 mmol/L glycine (pH 8.3) under conditions of 150 V and 50 mA for 90 min. Subsequently, the gel was incubated for 20 min shielded from light in a solution composed of 100 mL of phosphate buffer (pH 7.8), 0.098 mmol/L nitro blue tetrazolium, and 0.03 mmol/l vitamin B2. This incubation was then followed by a 30 min immersion of the gel in a solution containing 0.5 mL of TEMED (N, N, N, N-tetramethyl ethylenediamine) and 50 mL of water, in the absence of light. Finally, the gel was rinsed with distilled water and the bands were visualized in natural light [64].

### 4.4. Determination of Metabolic Enzyme Activities in Rat Colon Tissues

#### 4.4.1. Determination of Total LDH Activity

The enzymatic activity of LDH was gauged by observing the change in absorbance at 340 nm via a spectrophotometer. This absorbance change occurs due to NADH oxidation in the following equilibrium reaction: pyruvate + NADH + H+ ←→ lactate + NAD+ [65]. The assay was initiated by incorporating Na-pyruvate (0.1 mL, 23 mmol/L) to a reaction mixture containing 2.9 mL phosphate buffer (0.1 mM), pH = 7.0; NADH (0.05 mL, 14 mmol/L, which was freshly prepared and maintained on ice to prevent the spontaneous oxidation of NADH); and sample (supernatant of rat colon tissue homogenate, 0.01 mL). The definition of one unit of LDH activity (U) was the enzyme’s ability to convert 1 μmol of NADH each minute under the assay conditions. The total LDH activity, reflected by the reduction in absorbance, was quantified using a UV-2600 spectrophotometer (Shimadzu, Kyoto, Japan).

#### 4.4.2. Determination of Total MDH Activity

The assessment of MDH activity was conducted by monitoring the decrease in absorbance at 340 nm, which corresponds to the oxidation of NADH in the following reaction: oxalacetate + NADH + H+ ←→ malate + NAD+ at 340 nm [66]. The reaction was started by adding the substrate Na_2_-oxaloactetate (0.1 mL, 15 mmol/L) to the reaction mixture, which included 2.9 mL phosphate buffer (0.1 mM), pH = 7.5; NADH (0.05 mL, 14 mmol/L); and sample (supernatant of rat colon tissue homogenate, 0.01 mL). The definition of a single unit of MDH activity (U) is the conversion of 1 μmol of NADH each minute under the specified conditions. The cumulative MDH activity, reflected by the decrease in absorbance, was quantified utilizing a UV-2600 spectrophotometer (Shimadzu, Kyoto, Japan).

#### 4.4.3. Determination of LDH Isoforms

The distinct isoforms of LDH in the rat colon tissues were identified through direct electrophoretic zymography. Electrophoresis was performed on 7.5% native polyacrylamide gels and visualization of the isoforms was achieved by incubating the gel in a reaction mix comprising Li-lactate 0.25 g, NAD 18.75 mg, phenazinmethosulphate (PMS) 0.625 mg, nitrobluetetrazolium (NBT) 12.5 mg, and TRIS buffer 25 mL, pH 8.3. This incubation occurred for 10 min at a temperature of 37 °C in a dark environment. The LDH isoforms manifested as dark blue bands on the gel, indicative of formazan that forms through the reduction of NBT in the presence of the electron mediator PMS, with NAD acting as a coenzyme [67]. A quantitative analysis of the LDH isoforms on the resulting zymograms was performed with the ImageJ Q.42 software (Laboratory for Optical and Computational Instrumentation, University of Wisconsin, Madison, WI, USA). Their relative activities were quantified by measuring the area under the densitometric curves in pixels and the values were normalized to the protein concentration in the sample and are presented as px/mg protein.

#### 4.4.4. Determination of MDH Isoforms

The different isoforms of MDH in the rat colon tissues were determined by direct electrophoretic zymography. The electrophoresis process utilized a 7.5% native polyacrylamide gel. The isoforms were rendered visible by incubating the gel in a reaction solution containing Na2-malate 0.25 g, NAD 15 mg, PMS 0.5 mg, NBT 10 mg and phosphate buffer 25 mL, pH = 7.1, for 10 min at 37 °C in the absence of light. After visualization, the isoforms of MDH presented as dark blue bands on the gel, indicative of formazan. This is the end-product of NBT reduction, a product of the reduction of NBT in the presence of PMS as an electron transfer agent and NAD as a coenzyme [68].

### 4.5. Determination of Histomorphometric Parameters in Rat Colon Wall

Samples of the rat colon tissue were preserved in 4% neutral buffered formaldehyde solution for 48 h. Subsequent to fixation, the samples underwent dehydration and were then encased in paraplast. The embedded tissues were sectioned using a microtome (Leica Reinhart Austria and Leica SM2000 R, Heidelberg, Germany) producing slices with a thickness of 5 μm. These sections of the rat colon tissues were stained with hematoxylin and eosin. The following histomorphometric parameters were measured: the depth of the crypts, the thickness of the lamina muscularis mucosae, the thickness of the tunica mucosa, and the total colon wall thickness. An Olympus BX41 microscope (Tokyo, Japan) with an Olympus C5060-ADU camera (Tokyo, Japan) were used for the analysis. The histomorphometric parameters were quantified using the Image J Q.42 software (Laboratory for Optical and Computational Instrumentation, University of Wisconsin, Madison, WI, USA).

### 4.6. Statistical Analysis

Statistical analysis was performed using the SPSS 23.0 software package. The Shapiro–Wilk test was used for the determination of the distribution of data. The statistical comparisons between the experimental groups were conducted using ANOVA with Tukey’s post hoc test or the Kruskal–Wallis test followed by the Dunn–Bonferroni post hoc test where appropriate. Differences were considered significant at *p* < 0.05.

## 5. Conclusions

Subchronic treadmill training in rats with hyperhomocysteinemia triggers the colon tissue antioxidant response (by increasing the activities of SOD and CAT), changes colon morphometry parameters (induces a decrease in the crypt depth), and elicits an increase in metabolic enzyme activities (LDH and MDH). These results provide new insights into the effects of physical activity on the colon tissue in hyperhomocysteinemia.

The study offers a comprehensive assessment of the effects of aerobic exercise on colonic tissues in a rat model of hyperhomocysteinemia, evaluating a range of biological indicators including antioxidant enzyme activity, metabolic enzyme activity, and morphometric parameters, which suggest that exercise may confer protective effects at both the physiological and morphological levels. The results are statistically significant and provide practical insights into the inclusion of an appropriate regimen of physical activity to reduce HHcy as strategies to prevent inflammatory bowel disease and colorectal cancer.

## Figures and Tables

**Figure 1 ijms-25-01946-f001:**
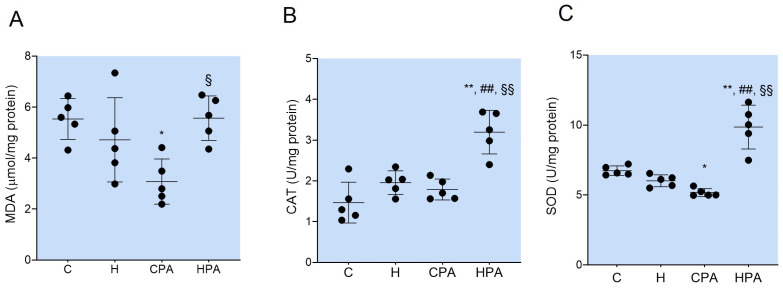
Concentration of malondialdehyde (MDA) (**A**), activity of catalase (CAT) (**B**), and total activity of superoxide dismutase (SOD) (**C**) in rat colon tissues. C—saline; H—homocysteine; CPA—saline + aerobic treadmill training; HPA—homocysteine + aerobic treadmill training. * *p* < 0.05 vs. C; ** *p* < 0.01 vs. C; ## *p* < 0.01 vs. H; § *p* < 0.05 vs. CPA; §§ *p* < 0.01 vs. CPA. One-way ANOVA with Tukey post hoc test.

**Figure 2 ijms-25-01946-f002:**
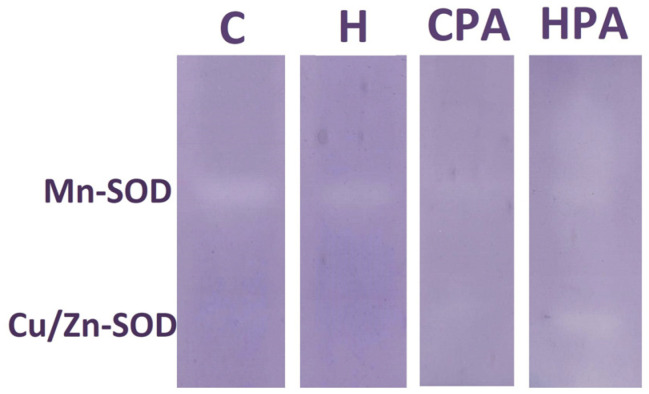
Representative zymograms of superoxide dismutase isoforms Mn-SOD and Cu/Zn-SOD in rat colon tissues. C—saline; H—homocysteine; CPA—saline + aerobic treadmill training; HPA—homocysteine + aerobic treadmill training. In the C, H and CPA groups, only Mn-SOD activity was detected while in the HPA group, besides the Mn-SOD isoform, Cu/Zn-SOD also showed activity.

**Figure 3 ijms-25-01946-f003:**
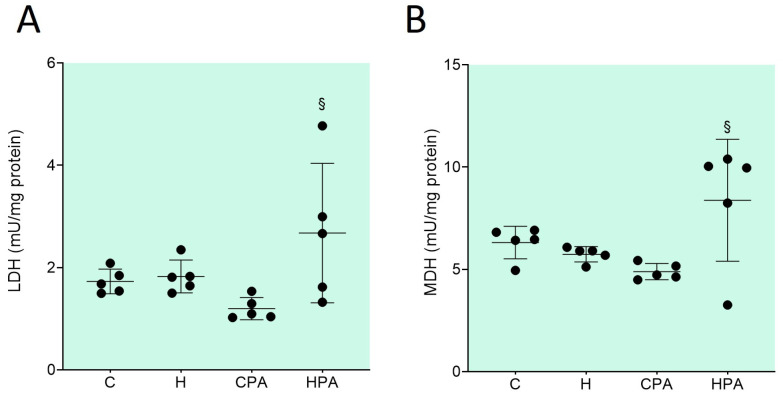
Total activity of lactate dehydrogenase—LDH (**A**) and malate dehydrogenase—MDH (**B**) in rat colon tissue. C—saline; H—homocysteine; CPA—saline + aerobic treadmill training; HPA—homocysteine + aerobic treadmill training. (**A**): § *p* < 0.05 vs. CPA. One-way ANOVA with Tukey post hoc test. (**B**): § *p* < 0.05 vs. CPA—Kruskal–Wallis with Dunn–Bonferroni post hoc test.

**Figure 4 ijms-25-01946-f004:**
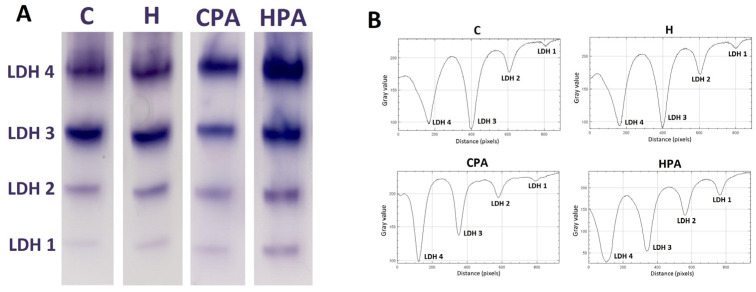
Lactate dehydrogenase in rat colon tissue by electrophoresis. Representative zymograms of lactate dehydrogenase isoforms (**A**) and representative densitometric curves of lactate dehydrogenase isoforms (**B**). Four LDH isoforms were detected: LDH 1, LDH 2, LDH 3, and LDH 4. C—saline; H—homocysteine; CPA—saline + aerobic treadmill training; HPA—homocysteine + aerobic treadmill training.

**Figure 5 ijms-25-01946-f005:**
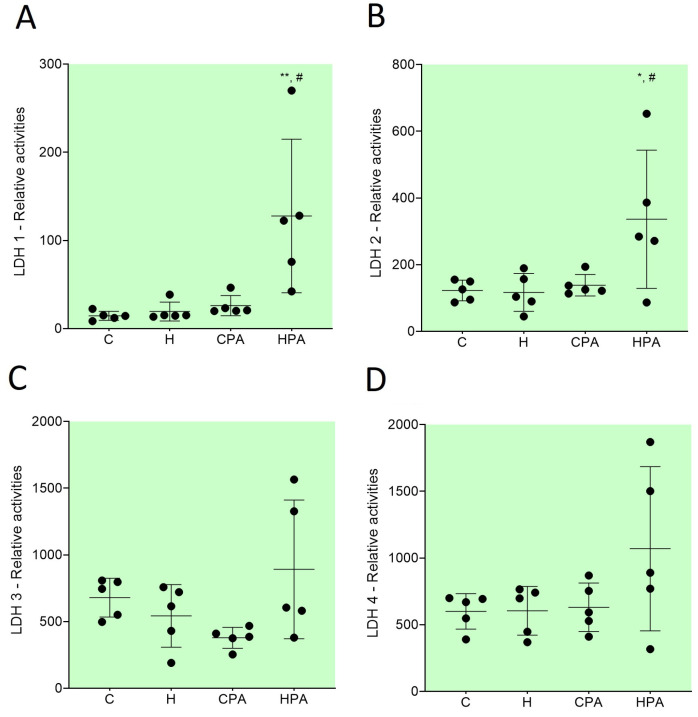
Relative activities of lactate dehydrogenase (LDH) isoforms in rat colon tissues: LDH 1 (**A**), LDH 2 (**B**), LDH 3 (**C**), LDH 4 (**D**). C—saline; H—homocysteine; CPA—saline + aerobic treadmill training; HPA—homocysteine + aerobic treadmill training. (**A**): ** *p* < 0.01 vs. C; # *p* < 0.05 vs. H (Kruskal–Wallis with Dunn–Bonferroni post hoc test). (**B**): * *p* < 0.05 vs. C; # *p* < 0.05 vs. H (one-way ANOVA with Tukey post hoc test).

**Figure 6 ijms-25-01946-f006:**
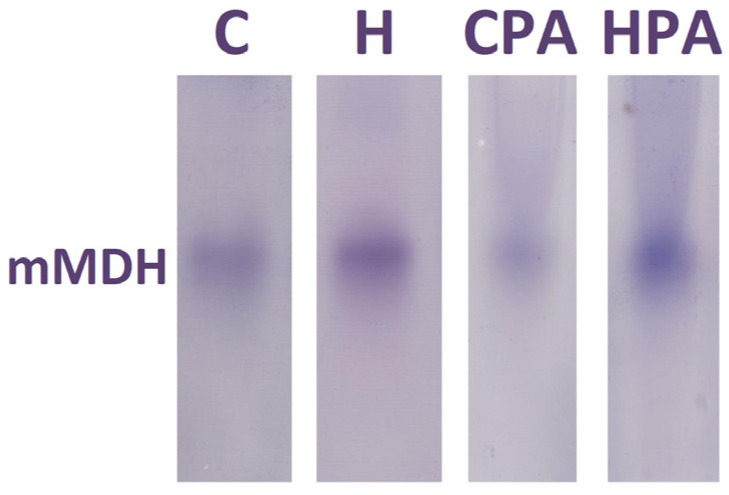
Representative zymograms of mitochondrial malate dehydrogenase MDH (mMDH) isoform in rat colon tissue. C—saline; H—homocysteine; CPA—saline + aerobic treadmill training; HPA—homocysteine + aerobic treadmill training.

**Figure 7 ijms-25-01946-f007:**
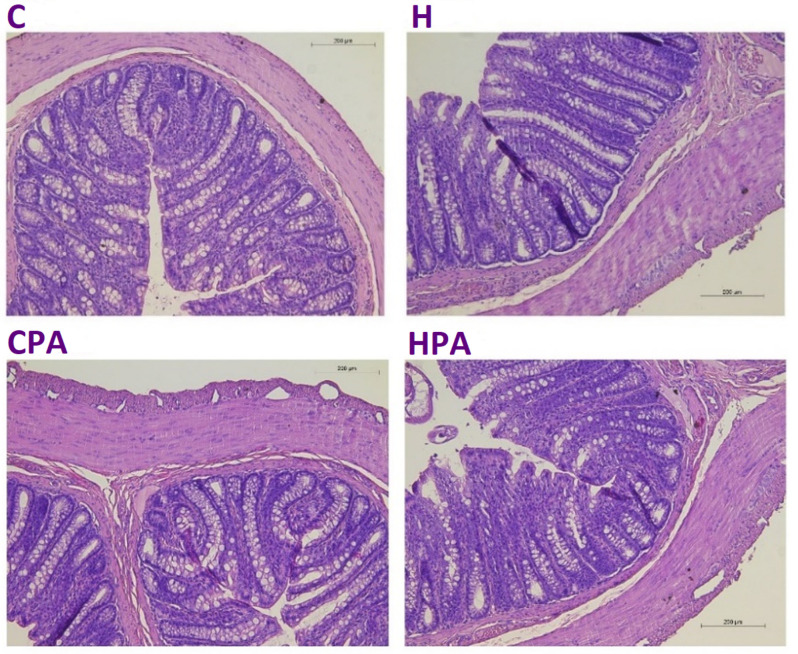
Cross-section of the colon with hematoxylin and eosin staining (magnification: ×100). C—saline; H—homocysteine; CPA—saline + aerobic treadmill training; HPA—homocysteine + aerobic treadmill training.

**Figure 8 ijms-25-01946-f008:**
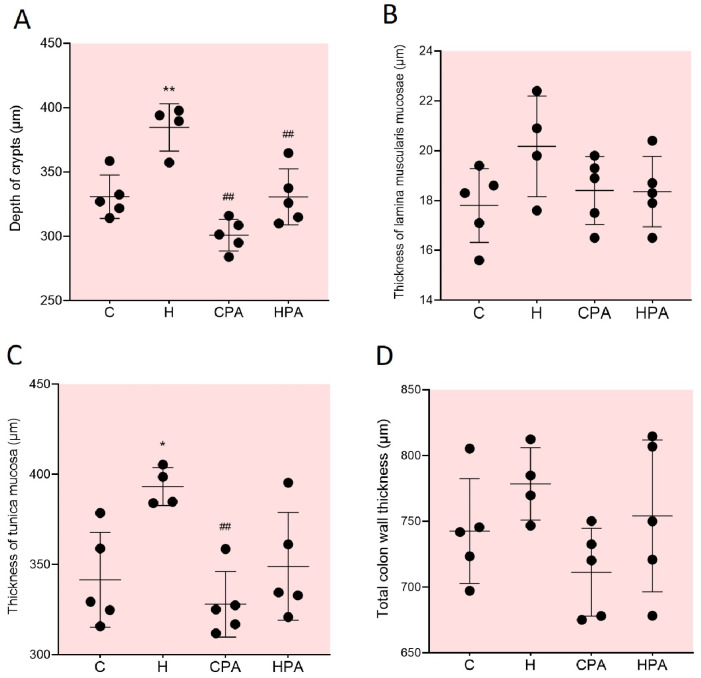
Histomorphometric parameters in rat colon tissues: depth of crypts (**A**), thickness of lamina muscularis mucosae (**B**), thickness of tunica mucosa (**C**), total colon wall thickness (**D**). C—saline; H—homocysteine; CPA—saline + aerobic treadmill training; HPA—homocysteine + aerobic treadmill training. * *p* < 0.05 vs. C; ** *p* < 0.01 vs. C; ## *p* < 0.01 vs. H. One-way ANOVA with Tukey post hoc test.

## Data Availability

The data presented in this study are available on request from the corresponding author. The data are not publicly available due to the highly specialized nature of the experimental protocols.

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
