# Peer review of "Effects of Aerobic Treadmill Training on Oxidative Stress Parameters, Metabolic Enzymes, and Histomorphometric Changes in Colon of Rats with Experimentally Induced Hyperhomocysteinemia"

_ijms, 2024, doi:10.3390/ijms25041946_

Round 1

Reviewer 1 Report

Comments and Suggestions for Authors

The manuscript offers a comprehensive assessment of the effects of aerobic exercise on colonic tissue in a rat model of hyperhomocysteinemia, evaluating a range of biological indicators including antioxidant enzyme activity, metabolic enzyme activity, and morphometric parameters, suggesting that exercise may confer protective effects at both physiological and morphological levels. The results are statistically significant and provide practical insights into the prevention of inflammatory bowel disease and colorectal cancer, as clearly stated by the authors.

However, there is a notable issue regarding the elucidation of the physiological mechanisms underlying the observed effects. The molecular pathways underpinning the exercise-induced increase in antioxidant enzyme activity, changes in metabolic enzyme activity, and effects on colonic morphometric parameters remain unclear. Clarifying these biological processes is crucial for designing intervention strategies to maximize the protective effects of exercise. Although it may be a future research task, it would be beneficial to include in the discussion some hypotheses on what physiological mechanisms could be hypothesized. Even if it is not possible to specify concrete molecular pathways at this stage, hypothesizing about potential signaling pathways or metabolic alterations would aid readers in more deeply interpreting the implications of the findings. Moreover, suggesting future research directions and methodologies on how exercise might impact cellular signaling, gene expression, and protein activation could expand the applicability of this research.

Author Response

Response to Reviewer 1 is uploaded as Word file

Reviewer 2 Report

Comments and Suggestions for Authors

[The significance of the research]

Hyperhomocysteinemia (HHcy) is a problem in contemporary society. It causes several diseases and the authors particulate especially in colorectal disease caused by HHcy. The authors also focused on the potential of training. This is both enlightening and practical.

[Title and Purpose]

The authors declare the purpose of this manuscript in the ABSTRACT section “The aim of this study was to investigate the effects of independent homocysteine load for two weeks, as well as its effects under the condition of aerobic treadmill training for four weeks on oxidative stress parameters, metabolic enzymes and histomorphometric changes in colon of rats with experimentally induced hyperhomocysteinemia.” This declaration should have been shorter and more simple. The purpose section became only a copy of the results, and maybe for the same reason, the authors failed to declare a concluding remark(s) in ABSTRACT.

[Logics]

The purpose of the authors and endpoint analyte is not connected in the Introduction and Result section in the present manuscript. For example, potential readers may have some difficulty understanding why the malondialdehyde CAT and SOD should be measured. The same points can be also said to be the other experiments. These points are related to radical part in the scientific writing, therefore, the reviewer decided to reject the article. However, the reviewer recommends resubmit after correction because the data in Figure 8 may lead attensions from the potential readers in IJMS.

In addition, the reviewers will point out individual points to publish this manuscript.

********************************************************************************

[Major points]

CuZn SOD, the total activity of LDH and MDH, LDH-isoforms, etc… ; In these experiments, only HPA was significantly or markedly different from other groups, even in the CPA group which only work out using a treadmill. Considering our general knowledge, the CPA group should have been different from the control group, but the data shows not. Does this mean Hcy is one of the conditions for up-regulation of these things? Is Hcy a necessary evil? Please explain in the discussion section.

Why do the authors compare H and CPA groups only in MDA and SOD concentration?

Why MDA and SOD levels were not up-regulated in the H group?

Authors should not just cherry-pick the most convenient data and explain it away.

Author Response

Response to Reviewer 2 is uploaded as Word file

Round 2

Reviewer 2 Report

Comments and Suggestions for Authors

[2md revision]

The author added some references and responded to the reviewer‘s question. The abstract was improved. Then, the reviewer will comment on your particular response here.

Comment 1

The authors insist in the first revision that (Response 3)

It is noted in the Materials and methods section that these parameters are measured in order to determinate the status of oxidative stress in colon tissue.

However, the potential readers usually read top to bottom of the article. Please explain carefully why these experiments were done before the experiments appear. Concretely, the author should introduce the fact that MDA, CAT, and SOD levels are representers of oxidative stress in the animals.

Moreover, the M&M section only describes what the method obtains, but the potential readers want to know why the authors need to select the method for what purpose. The present manuscript lacks logic in bridging the Introduction with the Result.

Also, the authors should obey the abbreviation rule. Abbreviations are listed when the word or phrase first appears.

Ex) Line 110 malondialdehyde -> malondialdehyde (MDA)

Line 115 catalase -> catalase (CAT)

Line 147 LDH -> lactate dehydrogenase (LDH)

And others

Comment 2

experimentally induced HHcy” should be “experimentally injected HHcy”, because the authors directly injected HHcy (from line 414), not by using stress inducers.

Author Response

Response to the Reviewer 2 comments is uploaded as PDF file.

Round 3

Reviewer 2 Report

Comments and Suggestions for Authors

The authors responded to the reviewer's comment properly. The manuscript became worth publishing.